



# Regional Terrestrial Water Storage Changes in the Yangtze River Delta over the Recent 20 years

Fengwei Wang[1, 2], Jianhua Geng[2*], Yunzhong Shen[3], Yanlin Wen[1, 4], Tengfei Feng[3]

[1] Shanghai Sheshan National Geophysical Observatory, Shanghai, PR, China
[2] State Key Laboratory of Marine Geology, Tongji University, Shanghai, PR, China
[3] College of Surveying and Geo-informatics, Tongji University, Shanghai, PR, China
[4] Shanghai Earthquake Agency, Shanghai, PR, China

* *Correspondence to*: Jianhua Geng (jhgeng@tongji.edu.cn)

**Abstract.** Monitoring changes in regional terrestrial water storage (TWS) and groundwater storage (GWS) is important for effectively managing water resources. Here, we investigate the TWS and GWS changes in the Yangtze River Delta using the GRACE/GRACE-FO mascon solutions, GLDAS NOAH models and in situ groundwater level changes from monitoring wells over the period of April 2002 to December 2022. The results show that the regional mean TWS change rate of the entire Yangtze River Delta is 0.62±0.10 mm/year, at 0.47±0.07 mm/year for the GWS component and 0.15±0.08 mm/year for the other components, which includes soil moisture, snow water and surface water change derived from the GLDAS NOAH models. At the basin scale, significant positive trends existed in water storage in Shanghai and Zhejiang Provinces; however, relatively small negative trends existed in Jiangsu and Anhui Provinces, which was confirmed by the spatial distributions of areas with linear trends. After comparing the estimated GWS change with the in situ groundwater level change from thirteen monitoring wells, we concluded that the groundwater levels in Shanghai and Zhejiang Provinces slightly recovered over the last 20 years and that this trend will continue in the coming years, mainly due to the sustainable water resource management policies of the local governments.

**Keywords.** Terrestrial Water Storage Change; GRACE/GRACE-FO; Yangtze River Delta

## 1 Introduction

Due to increasing global warming, there have been frequent extreme climatic events, which have had a great impact on the global water cycle and regional hydrological processes [Harder et al., 2015; Ummenhofer & Meehl, 2017; Yang et al., 2021]. Typically, changes in terrestrial water storage (TWS) are related to changes in soil moisture (SM), snow water equivalent (SWE) and surface water (SW) and groundwater storage (GWS) [Humphrey et al., 2023]. Many researchers have studied global and typical regional TWS or groundwater storage (GWS) changes using different datasets over various periods [Feng et al., 2013, 2022; Chen et al., 2020; Humphrey et al., 2023; Huan et al., 2023; Li et al., 2024]. In the early stage of this research, the TWS change was determined by computing the residual of other observed water fluxes within the hydrological budget, which could be derived based on the hydrological budget, assuming that the difference between precipitation, evapotranspiration and runoff is equivalent to the water storage change, which requires sufficiently accurate estimates of



precipitation, evapotranspiration and runoff fluxes [Petch et al., 2023]. For GWS changes, traditional methods, including pressure gauges, ground network measurements, and groundwater models, are often used to estimate global or regional GWS changes; however, these methods have substantial limitations for model establishment and data acquisition [Xie et al., 2018; 35 Feng et al., 2022; Liu et al., 2022; Huan et al., 2023].

Based on the development of satellite gravity technology, the Gravity Recovery and Climate Experiment (GRACE) gravity satellite was in operation from 2002 to 2017 [Tapley et al., 2019], and its successor, GRACE Follow-on (GRACE-FO), was also successfully launched in May 2018 [Landerer et al., 2020], which provides a direct way to estimate TWS change. Note that one can directly estimate the TWS component related to soil moisture, snow water and surface water changes by 40 GLDAS hydrological data. Therefore, with the emergence of satellite gravimetry combined with hydrological data, one can further investigate regional GWS changes by subtracting the TWS component derived from hydrological GLDAS models [Feng et al., 2022; Li et al., 2024].

The Yangtze River Delta (YRD), which includes Shanghai, Jiangsu, Zhejiang and Anhui Provinces, one of the most developed and affluent regions in China, is a key economic hub and plays a vital role in China's economy. In addition, urban 45 subsidence is a significant challenge for Shanghai, which may be due to various factors, such as groundwater extraction, construction activities, and geological conditions. To address this issue, measures such as groundwater pumping controls have been implemented in Shanghai. These efforts are crucial for mitigating the impacts of urban subsidence and ensuring the sustainable development of cities. To investigate TWS and GWS changes in a city such as Shanghai, we first analysed the TWS and GWS changes in the entire Yangtze River Delta using GRACE/GRACE-FO mascon solutions, GLDAS 50 NOAH models, and in situ groundwater level changes from monitoring wells from April 2002 to December 2022. In addition, we estimated the TWS and GWS changes for each province to study groundwater level changes to better formulate sustainable water resource management policies. The paper is organized as follows: the study area, employed datasets and processing methods are briefly described in Section 2. The results and analyses are presented in Section 3. Concluding remarks are given in Section 4.

## 55 2 Study area, employed datasets and processing methods

### 2.1 Study Area

The Yangtze River Delta, abbreviated as the YRD, includes Shanghai, Jiangsu Province, Zhejiang Province, and Anhui Province and is located in the lower reaches of the Yangtze River in China, adjacent to the Yellow Sea and the East China Sea. It is at a junction where the river meets the sea, with numerous ports along the river and the coast. It is a delta formed by 60 sediment deposition as the Yangtze River flows into the sea. The Yangtze River Delta mainly has a subtropical monsoon climate. The average annual temperature, as well as the average highest and lowest temperatures, have significantly increased in the study area over the last few years. The rate of warming is higher in winter and spring and lower in summer.

 

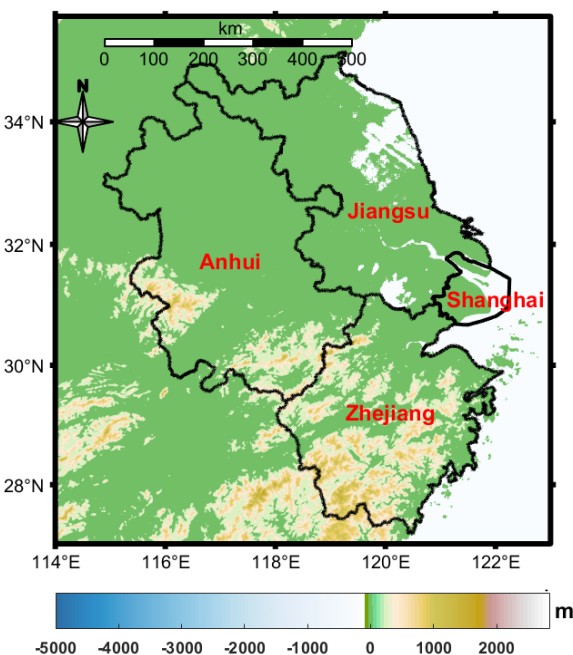

**Figure 1.** The study region of the Yangtze River Delta

## 2.2 GRACE and GRACE-FO Mascon Solutions

The TELLUS GRACE (2002.4-2017.6) and GRACE-FO (2018.6-2022.12) RL06 mascon solutions from the Center for Space Research (CSR) (Save et al., 2016; Save, 2020) and the Jet Propulsion Laboratory (JPL) (Watkins et al., 2015; Wiese et al., 2016) are jointly used to directly estimate the regional terrestrial water storage change in the Yangtze River delta from April 2002 to December 2022. Both the CSR and JPL mascon solutions are corrected for glacial isostatic adjustment (GIA) using the ICE6G-D model (Peltier et al., 2018). The C20 (degree 2 order 0) coefficients are replaced with the C20 solutions from the GRACE/GRACE-FO Technical Note 14 (Loomis et al., 2019). The degree-1 coefficient (Geocenter) corrections are applied using the estimates in GRACE/GRACE-FO Technical Note 13 (Landerer, 2019).

## 2.3 GLDAS NOAH Models

Hydrological models have often been used to validate the variations in the gravity field caused by variations in soil moisture, snow, surface water and other hydrological components across land regions. Here, we employ the GLDAS Noah model (GLDAS_NOAH10_M 2.1: Beaudoing & Rodell, 2020) with monthly temporal resolution and 1^o×1^o spatial resolution from April 2002 to December 2022, which can be downloaded from https://10.5067/LWTYSMP3VM5Z, without including the groundwater component. Note that all the employed datasets were interpolated to the same grid size and a 0.25° spatial resolution to better distinguish the boundaries.





**2.4 Groundwater Level Changes in Monitoring Wells**

Generally, to validate the reliability of the estimated GWS change in the Yangtze River Delta, the groundwater level changes from monitoring wells are used for comparison. In this study, thirteen monitoring wells located in the Yangtze River Delta are employed, including four wells in Shanghai (Chongming, Changxing Island, Fengcheng High School and Shanghai University), three wells in Zhejiang (Ningbo, Tongxiang, and Longyou), three wells in Anhui (Wuhe, Chaohu, and Anqing),

and three wells in Jiangsu (Guanyun, Jiangyan, and Nantong). Although groundwater level changes from monitoring wells may have relatively short available time spans with respect to the study period (2002.4~2022.12), the trend in groundwater level changes within a relative sub-period can also reflect GWS loss or increase to some extent.

**3 Changes in Terrestrial Water Storage in the Yangtze River Delta**

**3.1 Temporal Analysis**

To estimate the change in terrestrial water storage in the Yangtze River Delta, two GRACE/GRACE-FO mascon solutions are adopted. Because there were 33 missing months in the GRACE/GRACE-FO JPL mascon solutions from April 2002 to December 2020, particularly 11 consecutive missing months between the GRACE and GRACE-FO missions, the same missing months were deleted from the GLDAS NOAH data to maintain consistency and facilitate comparison. The estimated regional mean TWS changes in the Yangtze River Delta over the period from April 2002 to December 2022 are presented in

Figure 2. Figure 2 clearly shows that the estimates from CSR, JPL and GLDAS NOAH are consistent with each other, with some amplitude differences, especially for the smaller GLDAS NOAH, which may be mainly due to the missing groundwater change component. Note that for the estimated regional mean mass changes from the GRACE/GRACE-FO, GLDAS NOAH, and GRACE/GRACE-FO solutions minus GLDAS NOAH, the amplitudes of the annual and semiannual components and linear trends were coestimated by using the least-squares fitting approach, and the corresponding

uncertainties represent the least-squares fitting errors (1 sigma for the amplitudes, phases, linear trends). The statistical results are presented in Table 1. The mean regional mean TWS change rate in the Yangtze River Delta Basin is 0.62±0.19 mm/year according to the GRACE/GRACE-FO solutions, including 0.15±0.16 mm/year according to the GLDAS NOAH estimates, and the groundwater storage (GWS) change rate is 0.47±0.14 mm/year, indicating that GWS change dominates the TWS change in the Yangtze River Delta.





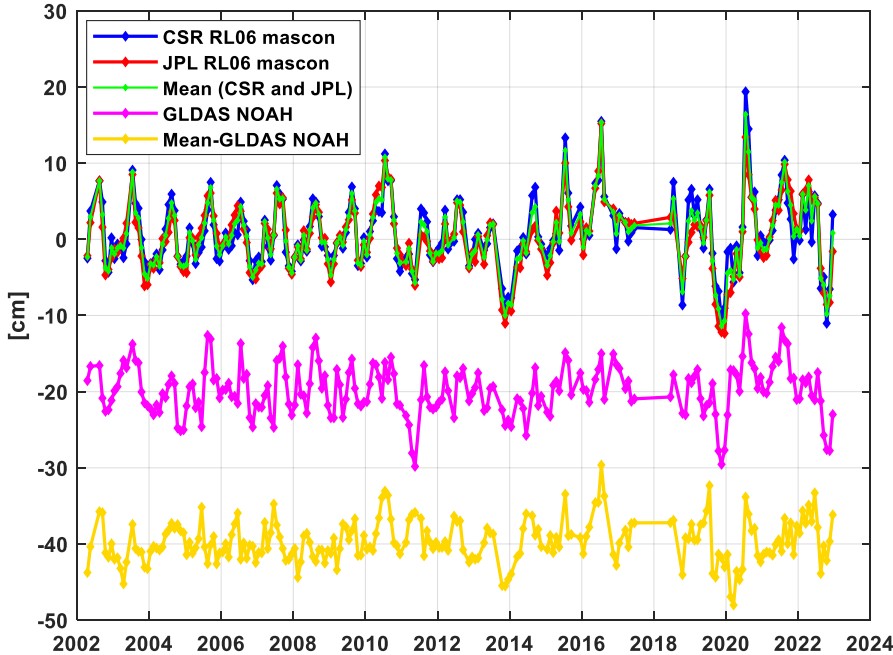

**Figure 2. Changes in terrestrial water storage and its components in the Yangtze River Delta from April 2002 to December 2022. Graphs are shifted along the ordinate axis to better show the data.**

**Table 1. Amplitudes of annual and semiannual components and linear trends in regional terrestrial water storage changes from GRACE/GRACE-FO and GLDAS Noah data for the period from April 2002 to December 2022**

| Type | Index | Annual Amplitude [cm] Phase [deg] | Semiannual Amplitude [cm] Phase [deg] | Linear Trend [mm/year] |
|---|---|---|---|---|
| TWS | CSR Mascon | [3.24±0.33] [192.6±5.8] | [2.55±0.33] [71.4±7.5] | 0.72±0.10 |
| | JPL Mascon | [3.50±0.35] [181.7±5.7] | [1.39±0.35] [76.2±14.5] | 0.51±0.10 |
| | Mean | [3.36±0.33] [186.9±5.6] | [1.97±0.33] [73.1±9.6] | 0.62±0.10 |
| SM+SWE+SW | GLDAS Noah | [1.54±0.28] [194.4±10.4] | [1.74±0.28] [95.2±9.2] | 0.15±0.08 |
| GWS | Mean-Noah | [1.84±0.23] [180.7±7.3] | [0.75±0.23] [11.4±17.8] | 0.47±0.07 |

*Note*. The uncertainty represents the least-squares fitting error (1 sigma for amplitudes, phases and trends). SM, SWE and SW represent soil moisture, snow water equivalent and surface water change, respectively.

In addition, to demonstrate the regional mean TWS and GWS changes in four basins, particularly in Shanghai city, Jiangsu, Anhui and Zhejiang Provinces, we estimated the corresponding basin mean TWS and GWS changes from GRACE/GRACE-FO and GLDAS NOAH over the period from April 2002 to December 2022. The TWS change and its components for Anhui, Jiangsu, and Zhejiang Provinces and Shanghai city are shown in Figure 3. The statistical results are presented in Table 2. Table 2 shows that the mean TWS change rates in Anhui and Jiangsu Provinces are -0.50±0.22 mm/year and -0.82±0.17





mm/year, respectively, indicating that slight mass loss occurred in these two basins, similar to the GWS change rate. However, for Shanghai and Zhejiang Provinces, the GWS change rates are 1.93±0.16 mm/year and 3.02±0.19 mm/year, respectively, indicating that the groundwater storage level in these two basins has recovered over the last 20 years and that this trend will continue in the coming years.

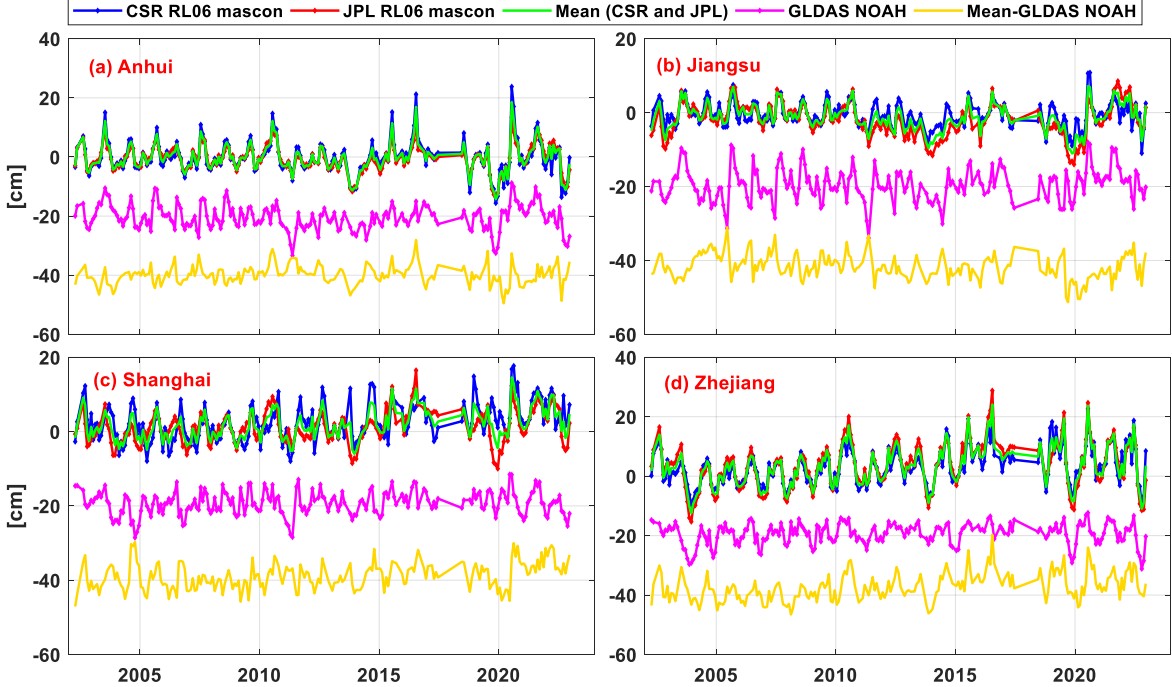

**Figure 3. Changes in terrestrial water storage and its components in Anhui, Jiangsu, and Zhejiang Provinces and Shanghai city. Graphs are shifted along the ordinate axis to better show the data.**

**Table 2. Amplitudes of annual and semiannual components and linear trends in regional terrestrial water storage change for Anhui, Jiangsu, Zhejiang and Shanghai from GRACE/GRACE-FO, meteorological and GLDAS Noah data for the period from April 2002 to December 2022.**

| Basin | Type | Index | Annual Amplitude [cm] Phase [deg] | Semi-annual Amplitude [cm] Phase [deg] | Linear Trend [mm/year] |
|-------|------|-------|-----------------------------------|----------------------------------------|------------------------|
| Anhui | TWS | CSR Mascon | [3.87±0.42] [193.1±6.3] | [3.05±0.42] [67.9±8.0] | -0.52±0.12 |
|  |  | JPL Mascon | [3.00±0.35] [185.6±6.7] | [1.57±0.35] [75.6±12.9] | -0.48±0.10 |
|  |  | Mean | [3.43±0.37] [189.8±6.3] | [2.31±0.37] [70.5±9.4] | -0.50±0.11 |
|  | SM+SWE+SW | GLDAS NOAH | [1.78±0.36] [206.7±11.6] | [2.07±0.36] [94.9±10.0] | -0.57±0.10 |
|  | GWS | Mean - GLDAS NOAH | [1.80±0.28] [173.1±8.9] | [0.95±0.28] [6.5±16.7] | 0.07±0.08 |
|  |  |  |  |  |  |
| Jiangsu | TWS | CSR Mascon | [1.92±0.24] [230.5±8.3] | [2.53±0.24] [83.2±5.5] | -0.70±0.07 |





| | | | | | |
|---|---|---|---|---|---|
| | | JPL Mascon | [1.68±0.40] [211.3±13.4] | [1.18±0.39] [100.3±19.2] | -0.95±0.12 |
| | | Mean | [1.77±0.30] [221.5±7.3] | [1.84±0.29] [88.7±9.2] | -0.82±0.08 |
| | SM+SWE+SW | GLDAS NOAH | [2.25±0.34] [264.4±8.7] | [2.94±0.34] [102.5±6.7] | 0.78±0.10 |
| | GWS | Mean - GLDAS NOAH | [1.54±0.29] [136.3±4.4] | [1.23±0.29] [303.3±13.5] | -1.60±0.08 |
| | | | | | |
| Shanghai | TWS | CSR Mascon | [3.89±0.33] [254.7±4.8] | [2.06±0.33] [111.6±9.2] | 3.29±0.10 |
| | | JPL Mascon | [3.38±0.32] [178.1±5.5] | [1.01±0.32] [79.7±18.5] | 1.91±0.10 |
| | | Mean | [2.86±0.28] [219.6±5.0] | [1.49±0.28] [101.2±10.7] | 2.60±0.08 |
| | SM+SWE+SW | GLDAS NOAH | [1.94±0.26] [171.7±7.9] | [1.05±0.26] [74.8±14.5] | 0.67±0.08 |
| | GWS | Mean - GLDAS NOAH | [2.12±0.29] [262.2±7.8] | [0.72±0.29] [142.0±21.9] | 1.93±0.08 |
| | | | | | |
| Zhejiang | TWS | CSR Mascon | [4.58±0.44] [172.7±5.5] | [2.06±0.44] [60.0±12.2] | 3.70±0.12 |
| | | JPL Mascon | [6.72±0.52] [171.3±4.4] | [1.62±0.52] [57.5±18.5] | 3.43±0.12 |
| | | Mean | [5.65±0.45] [171.9±4.6] | [1.84±0.45] [58.9±14.2] | 3.57±0.13 |
| | SM+SWE+SW | GLDAS NOAH | [2.97±0.28] [144.7±5.3] | [0.44±0.28] [58.0±36.4] | 0.54±0.13 |
| | GWS | Mean - GLDAS NOAH | [3.30±0.32] [196.1±5.6] | [1.40±0.32] [59.2±13.4] | 3.02±0.10 |

*Note.* The uncertainty represents the least-squares fitting error (1 sigma for amplitudes, phases and trends). SM, SWE and SW represent soil moisture, snow water equivalent and surface water change, respectively.

## 3.2 Spatial analysis

In Section 3.1, we investigated the regional TWS and GWS changes in the Yangtze River Delta at the temporal scale. To further analyse the spatial distribution of TWS and GWS in the Yangtze River Delta, we show the spatial distributions of linear trend and annual and semiannual amplitudes in Figure 4. Figure 4 shows that the northern Yangtze River Delta has obvious negative TWS change rates. When the TWS component derived from GLDAS NOAH is removed, the GWS change rate shows similar spatial distribution patterns. Note that for Shanghai city, and Zhejiang Province, the TWS and GWS

change rates have close to positive values, mainly because of the groundwater resource protection policies of local governments such as that of Shanghai, where mainly surface water is consumed with no groundwater extraction. In addition to human activities such as agricultural irrigation and industrial development, fossil resource mining leads to significant water consumption, especially coal mining. However, for the spatial distributions in terms of annual and semiannual amplitudes, there are some significant differences between the estimated TWS and GWS changes.




To further validate the estimated TWS and GWS change rates, the time series of thirteen well water level changes in the Yangtze River Delta were employed to test the reliability of the estimated GWS change rates at the given spatial scale. Note that the water level is the distance from the wellhead to the water surface; therefore, a decrease in the water level indicates a rise in the groundwater level. For example, for four wells in Shanghai (Chongming, Changxing Island, Fengcheng High School and Shanghai University), all the time series of the four water wells exhibited an increasing trend, indicating that the

groundwater storage level in Shanghai may have recovered due to groundwater recharge, similar to that in Zhejiang Province. Therefore, we believe that the TWS change rates in Shanghai and Zhejiang Provinces are mainly due to GWS changes. According to our comprehensive investigation, the negative trends in northern Anhui and Jiangsu Provinces may be mainly due to coal mining.

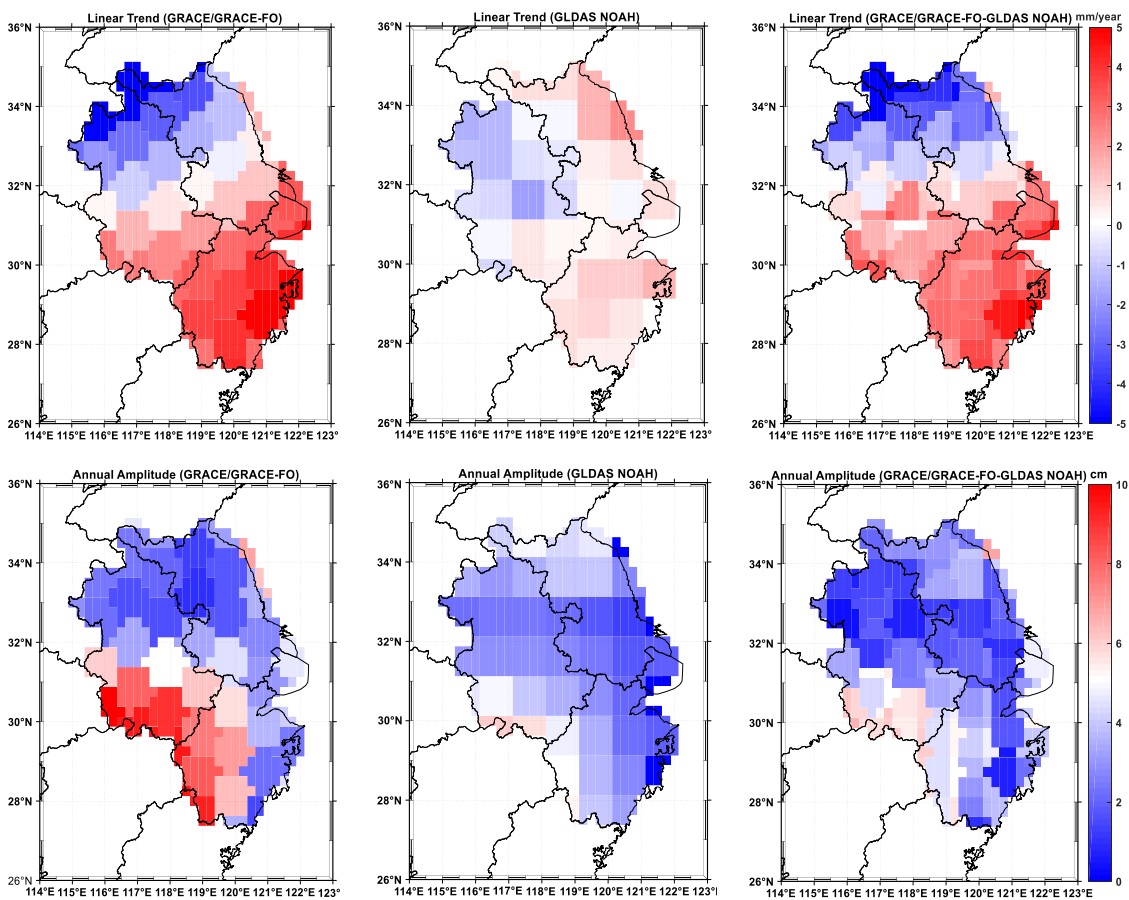



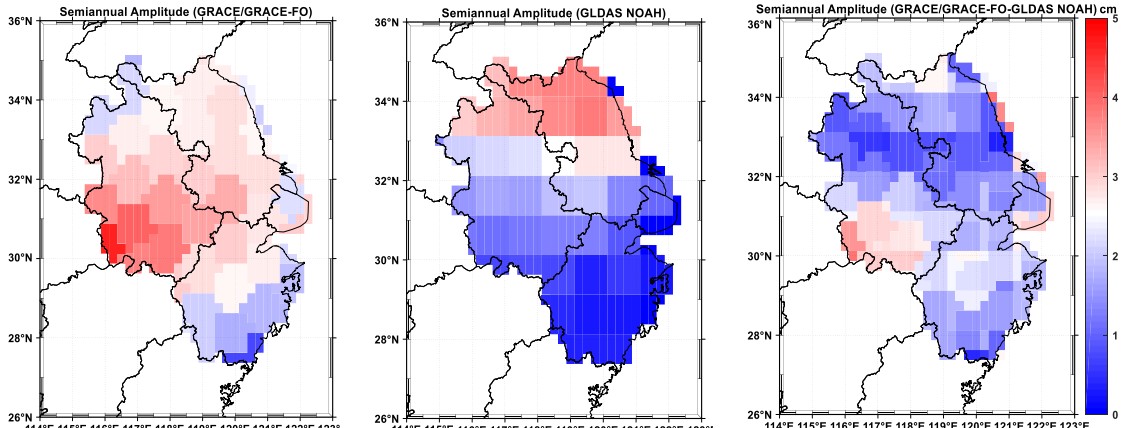

**Figure 4. The spatial distributions of linear trend (Row 1), annual (Row 2) and semiannual (Row 3) amplitudes from GRACE/GRACE-FO, GLDAS NOAH and its inferred estimates GRACE/GRACE-FO-GLDAS NOAH**

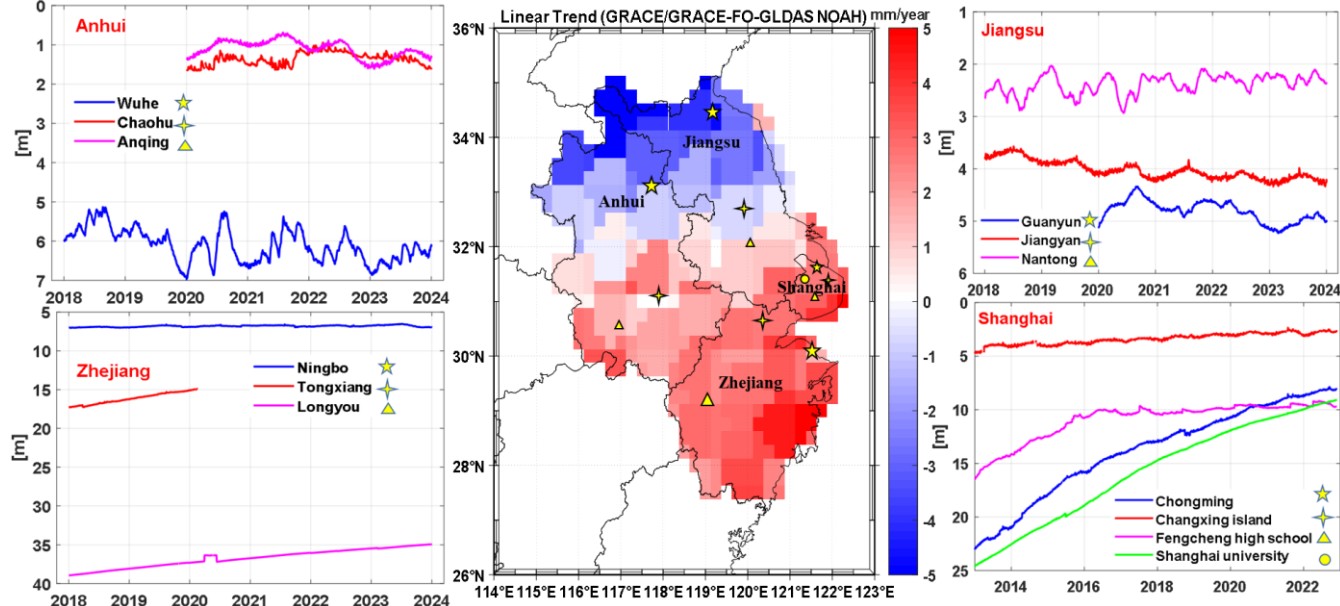

**Figure 5. The time series of corresponding well water level changes in the Yangtze River Delta. Note. Graphs are shifted along the ordinate axis to better show the data.**

## 4 Conclusions

In this study, the terrestrial water storage (TWS) and groundwater storage (GWS) in the Yangtze River Delta were estimated using multisource datasets for the period from April 2002 to December 2022. The analysed results show that the regional mean TWS and GWS change rates of the whole Yangtze River Delta are 0.62±0.10 mm/year and 0.47±0.07 mm/year, respectively, indicating that GWS change dominates the TWS change in the Yangtze River Delta. After analysing and



estimating the TWS and GWS changes in Shanghai, Zhejiang, Anhui and Jiangsu Provinces in detail, it was inferred that the

160 soil moisture, snow water equivalent change and surface water change dominate the TWS change in Anhui Province; however, GWS change dominates in Shanghai city and Zhejiang Province. In addition, for Jiangsu Province, the GWS change fully accounts for the TWS loss, which can be confirmed by the in situ groundwater level change series to some extent.

**Author Contributions:** Conceptualization, F.W. and J.G.; methodology, F.W. and F.T.; validation, F.W., and Y.W.;
investigation, Y.W.; resources, Y.W.; writing—original draft preparation, F.W.; writing—review and editing, F.W. and J.G.; supervision, Y.S.; funding acquisition, F.W.

**Funding:** This work is funded by the Shanghai Sheshan National Geophysical Observatory (SSKP202201) and Natural Science Foundation of China (42274005).

**Data Availability Statement:** Two GRACE/GRACE-FO (i.e. CSR and JPL) mascon solutions are available at the websites
from https://doi.org/10.15781/cgq9-nh24 and https://podaactools.jpl.nasa.gov/drive/files/all Data/tellus/L3/mascon/RL06, respectively. The GLDAS Noah models can be directly downloaded at (https://doi.org/10.5067/FOUXNLXFAZNY).

**Acknowledgments:** We acknowledge the CSR and JPL for providing the mascon solutions.

**Conflicts of Interest:** The authors declare no conflicts of interest.

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
