# Peer review of "Regional Terrestrial Water Storage Changes in the Yangtze River Delta over the Recent 20 years"

_EGUsphere, 2024_

## Referee Comment (RC2)

These comments are for "Regional Terrestrial Water Storage Changes in the Yangtze River Delta over the Recent 20 years" by Dr. Fengwei Wang. They investigate the terrestrial water and groundwater storage changes in the Yangtze River Delta. I think this manuscript lacks of discussion section, but this issue should be evaluated by Editor.

1) What is the white region in the north Jiangsu Province. Please explain this. What is the area of your study region? And the areas of four provinces. How do you consider the impact of small area in Shanghai City.

[Figure]

2) L76, check '1^o×1^o spatial'.

3) The link in L77 'https://10.5067/LWTYSMP3VM5Z' cannot open. It is different from the Data Availability Statement.

4) Please specifically state the unit of linear trends (mm/year) in Table 1 are different from amplitudes (cm) in the table caption. Since the unit of TWSA is cm, and the unit of TWSA trend is mm/year.

5) The extraction of semiannual amplitude should contain larger uncertainty when comparing the two solutions.

6) L111, how do you obtain the surface water change estimates in the GLDAS Noah? Please clarify this.

7) L119, the increased GWS trends can not be stated as 'recovered', since you do not know the GWS trends before 2002.

8) Figure 5, it is better to compare the GWS trends estimates for the periods of 2018-2023 or 2018-2022. These groundwater levels are groundwater depth? The groundwater depth is the distance from the ground surface. In addition, what is sources of the groundwater wells data, the authors should clarify this.